# Nutritional Interventions with *Bacillus coagulans* Improved Glucose Metabolism and Hyperinsulinemia in Mice with Acute Intermittent Porphyria

**DOI:** 10.3390/ijms241511938

**Published:** 2023-07-26

**Authors:** Miriam Longo, Daniel Jericó, Karol M. Córdoba, José Ignacio Riezu-Boj, Raquel Urtasun, Isabel Solares, Ana Sampedro, María Collantes, Ivan Peñuelas, María Jesús Moreno-Aliaga, Matías A. Ávila, Elena Di Pierro, Miguel Barajas, Fermín I. Milagro, Paola Dongiovanni, Antonio Fontanellas

**Affiliations:** 1Hepatology: Porphyrias & Carcinogenesis Laboratory, Solid Tumors Program, CIMA-University of Navarra, 31008 Pamplona, Spain; longo.miriam92@gmail.com (M.L.); djerico@alumni.unav.es (D.J.); kcordoba@alumni.unav.es (K.M.C.); asampedro@unav.es (A.S.); maavila@unav.es (M.A.Á.); 2Medicine and Metabolic Diseases, Fondazione IRCCS Ca’ Granda Ospedale Maggiore Policlinico, 20122 Milan, Italy; elena.dipierro@unimi.it (E.D.P.); paola.dongiovanni@policlinico.mi.it (P.D.); 3Center for Nutrition Research, Department of Nutrition, Food Sciences and Physiology, Faculty of Pharmacy and Nutrition, University of Navarra, 31008 Pamplona, Spain; jiriezu@unav.es (J.I.R.-B.); mjmoreno@unav.es (M.J.M.-A.); fmilagro@unav.es (F.I.M.); 4Navarra Institute for Health Research (IdiSNA), 31008 Pamplona, Spain; mcollant@unav.es (M.C.); ipenuelas@unav.es (I.P.); 5Biochemistry Area, Department of Health Science, Public University of Navarre, 31008 Pamplona, Spain; raquel.urtasun@unavarra.es (R.U.); miguel.barajas@unavarra.es (M.B.); 6Rare Disease Unit, Internal Medicine Department, Clinica Universidad de Navarra, 31008 Pamplona, Spain; isolares@unav.es; 7MicroPET Research Unit, CIMA-CUN, 31008 Pamplona, Spain; 8Nuclear Medicine-Department, CUN, 31008 Pamplona, Spain; 9Centro de Investigación Biomédica en Red de Fisiopatología de la Obesidad y Nutrición (CIBEROBN), Instituto de Salud Carlos III, 28029 Madrid, Spain; 10Centro de Investigación Biomédica en Red de Enfermedades Hepáticas y Digestivas (CIBERehd), Instituto de Salud Carlos III, 28029 Madrid, Spain

**Keywords:** glucose homeostasis, gut microbiome, hepatic porphyria, insulin resistance, metabolic disease, nutritional intervention, probiotics

## Abstract

Acute intermittent porphyria (AIP) is a metabolic disorder caused by mutations in the porphobilinogen deaminase (PBGD) gene, encoding the third enzyme of the heme synthesis pathway. Although AIP is characterized by low clinical penetrance (~1% of PBGD mutation carriers), patients with clinically stable disease report chronic symptoms and frequently show insulin resistance. This study aimed to evaluate the beneficial impact of nutritional interventions on correct carbohydrate dysfunctions in a mouse model of AIP that reproduces insulin resistance and altered glucose metabolism. The addition of spores of *Bacillus coagulans* in drinking water for 12 weeks modified the gut microbiome composition in AIP mice, ameliorated glucose tolerance and hyperinsulinemia, and stimulated fat disposal in adipose tissue. Lipid breakdown may be mediated by muscles burning energy and heat dissipation by brown adipose tissue, resulting in a loss of fatty tissue and improved lean/fat tissue ratio. Probiotic supplementation also improved muscle glucose uptake, as measured using Positron Emission Tomography (PET) analysis. In conclusion, these data provide a proof of concept that probiotics, as a dietary intervention in AIP, induce relevant changes in intestinal bacteria composition and improve glucose uptake and muscular energy utilization. Probiotics may offer a safe, efficient, and cost-effective option to manage people with insulin resistance associated with AIP.

## 1. Introduction

Acute intermittent porphyria (AIP) is an autosomal dominant disorder characterized by a relatively high frequency of porphobilinogen deaminase (PBGD) gene mutations (1/1700 of newborns) [1,2] but low and incomplete clinical penetrance, encompassing ~1% of PBGD mutation carriers [3,4]. Although PBGD haploinsufficiency is at the core of the onset, upregulation of the first rate-limiting enzyme of the pathway, 5-aminolevulinic acid (ALA) synthase-1 (ALAS1), induces hepatic overproduction and a consequent accumulation of ALA and porphobilinogen (PBG) in plasma and urine. These heme precursors are associated with the complex set of neurotoxic manifestations exhibited by patients with AIP [5]. Chronic abdominal pain, gastrointestinal symptoms (abdominal pain, vomiting, nausea, and constipation), tachycardia, and hypertension are the major manifestations associated with these acute attacks. Peripheral motor neuropathy and central nervous system disturbances (seizures, weakness, insomnia, hallucinations, and confusion) are rare and occur only during severe and commonly prolonged attacks. Long-term complications in AIP include chronic pain, non-epicirrhotic hepatocellular carcinoma and nephropathy, thereby representing a long-life burden for patients and relatives [6,7,8,9].

Acute attacks are associated with known trigger factors, including caloric deprivation and rapid weight loss, steroid hormones, medications and other chemicals, infections, stress and excess alcohol intake, which lead to the activation of hepatic *ALAS1* transcription (see network resources of drugs database and acute porphyrias: “http://www.porphyriafoundation.com/drug-database (accessed on 22 July 2023))”. Thus, disease stabilization consists of avoiding exposure to precipitating factors. For instance, AIP patients at-risk should not restrict calories, except for short periods and under the supervision of specialists. In women, the acute attack is also commonly related to fluctuations in sex-hormones. Premenstrum is the most common triggering factor, and when the bleeding starts, estrogen and progesterone levels decrease dramatically, normalizing many liver metabolic pathways which have been up- or down-regulated. The most common therapeutic option to manage and/or prevent the acute episodes consists of hemin infusion. In the absence of hemin or in mild cases, carbohydrate loading could be alternatively considered. Although its use in controlled trials has led to inconclusive results, some authors suggest that high sugar or fruit juice intake alongside hyperinsulinemia is correlated with less disease activity. Thus, a well-balanced diet has been recommended in AIP subjects in order to prevent the symptomatology, and high carbohydrate intake or glucose infusions have been included for the management of minor symptoms [10,11,12].

Given that most of patients remain asymptomatic throughout their lives (latent AIP), even if some show increased excretion of early heme precursors, they are not eligible to receive any therapeutic treatment. However, a recent study showed that 46.4% of them report chronic symptoms associated with porphyria, such as abdominal pain, fatigue, muscle pain, and insomnia [13]. In recent years, it has emerged that AIP subjects with stable disease have a higher prevalence of hyperinsulinemia and insulin resistance (IR) compared to either control volunteers, including family members without PBGD gene mutations and porphyrin accumulation, or refractory AIP patients [10,14].

Paradoxically, hyperinsulinemia downregulates hepatic *ALAS1* expression and may appear protective against the recurrence of acute attacks [10,14]. Moreover, IR may limit the efficacy of carbohydrate loading for the treatment of mild AIP cases [10,15,16,17].

The characterization of carbohydrate metabolic profiles in experimental models has highlighted alterations in glucose metabolism, hyperinsulinemia, and abnormal hepatic glycogenolysis, gluconeogenesis and mitochondrial bioenergetics that could be associated with AIP pathophysiology [10,16,17]. All these findings suggest that AIP may be considered a genetic disorder characterized by a miscellaneous set of metabolic dysfunctions.

Accumulating evidence has demonstrated that the gut microbiota regulates host metabolism, such as diabetes, obesity, fatty liver disease, and enteropathy, suggesting a potential role for nutritional intervention to improve insulin sensitivity and glycemic control in individuals with metabolic diseases [18,19]. The potential efficacy of probiotics, the dietary consumption of fermented foods, has been proposed as an effective approach to achieve health benefits for common metabolic disorders [20,21]. Probiotics are generally safe and well tolerated, with few side effects and the potential to improve insulin sensitivity and blood sugar control in a cost-effective manner. Thus, probiotic- and postbiotic-based diets appear as an attractive option for the management of patients with nonalcoholic fatty live disease (NAFLD), diabetes, and metabolic syndrome [22,23,24].

*Bacillus coagulans* (*B. coagulans*) is a Gram-positive bacterium belonging to the genus *Bacillus* that does not exist in the intestinal microbiota. Several studies have demonstrated that supplementing regular food with spores of *B. coagulans* (BC30™) enhances glucose tolerance and has the ability to maintain the health of intestinal bacteria [25,26,27,28,29,30]. BC30™ is currently one of the commonly used probiotic strains, with the advantages of high temperature resistance, high stability, and acid and alkali resistance [31]. Thus, due to these characteristics, it is a suitable choice as a highly stable spore-forming probiotic ingredient that supports digestive and immune health and protein absorption.

Neither probiotic- nor postbiotic-based diets have been tested for the management of AIP in previous studies. Given that they could be a viable treatment option for people with IR and overweight unable to tolerate fasting or medications due to their side effects, the aim of this work was to investigate whether nutritional interventions using *B. coagulans* spores can normalize metabolic alterations in a mouse model of AIP. We particularly focused on glucose metabolism and insulin response, looking at the liver and insulin-sensitive tissues such as muscle, adipose tissue, and the brain.

## 2. Results

### 2.1. B. coagulans Administration to AIP Mice Results in a Sustained Response to High Glucose Overload and Hyperinsulinemia

Specific parameters related to glucose metabolism were investigated in AIP mice fed alive probiotic *B. coagulans* for 12 weeks. The first parameter we focused on was the response to i.p. glucose overload. We carried out two types of GTT tests consisting of moderate-to-high glucose dose (2 g/kg and 5 g/kg, respectively) to investigate whether the treatments improved glucose tolerance in AIP mice and whether their efficacy persisted in a condition of severe hyperglycemia (≥300 mg/dL). In the GTT with 2 g/kg, AIP mice supplemented with *B. coagulans* showed a nonstatistically significant reduction in the total peak area under the curve (AUC) compared to the control AIP (Appendix A). As expected, severe hyperglycemia arose after increasing the glucose dose to 5 g/Kg (median blood glucose levels: 420.25 mg/dL IQR [389.37–474.87], measured at the peak ranging from 25 to 45 min after i.p. glucose injection, Figure 1A). The AUC, calculated from the first to the third hour post-glucose overload (at 5 g/kg i.p.), confirmed that in AIP mice, the reduction in hyperglycemia was delayed compared to WT animals (Figure 1B). At such a high concentration, AIP mice supplemented with *B. coagulans* showed a dramatic decline in the glucose peak after 30 min, as indicated by a steep slope from 30 to 90 min (Figure 1C). As a consequence, AIP mice supplemented with *B. coagulans* restored basal glycemia values at the same time as the control WT.

Higher serum insulin was detected in AIP mice (Figure 1D). The paradoxically high insulin levels observed in control AIP mice during fasting suggest that AIP pathogenesis may be associated with abnormal insulin secretion. AIP mice receiving *B. coagulans* restored serum insulin to the same levels as those found in fasted WT mice. These data show that a nutritional intervention with this probiotic is effective in improving both glycemic and insulinemic indexes in an AIP model.

### 2.2. B. coagulans-Supplemented Diet Reduced Body Fat Fraction and Increased Lean Mass in AIP Mice

Throughout the experimental trial, we monitored weekly food intake and body weight (BW) to evaluate whether *B. coagulans* supplementation modified nutritional habits and BW gain. While the median weekly ingestion per group was the same between the WT and AIP control animals (103.05 ± 9.98 g vs. 107.86 ± 16.02 g, respectively), mice fed *B. coagulans* showed the lowest food intake (82.74 ± 11.11 g, *p* < 0.01 vs. AIP), probably due to the satiety effect and/or the poor palatability of the diet. However, weight gain measured as g of BW gained (or lost)/kg of food ingested per week was similar between the two groups of AIP mice, either fed *B. coagulans* (*p* = 0.27; Figure 2A) or not. Basically, animals lose weight when energy consumption is higher than the food ingested, but body fat and lean tissue loss is often concurrently observed during extended caloric restriction [32]. In order to investigate whether weight loss was associated with changes in the body composition, precise fat and lean mass were measured using quantitative magnetic resonance imaging (MRI) in the alive animals at the end of the study. While no difference in both fat and lean/fat ratio were found among the control AIP and WT groups, the largest effect on fat composition was observed in AIP mice supplemented with *B. coagulans* (Figure 2B,C). The supplemented mice’s body fat reduced by 30% (Figure 2B) and concomitantly showed an increased lean/fat ratio of about 46% compared to the AIP mice (Figure 2C). These data indicate that reduced food intake in AIP mice supplemented with *B. coagulans* was associated with preserving or increasing lean mass.

### 2.3. The B. coagulans-Supplemented Diet Enhanced Glucose Uptake in the Skeletal Muscle, but Only Slightly Modified Glucose Uptake in the Liver and the Brain

By combining ex vivo measurements of [^18^F]Fluoro-2-deoxy-2-D-glucose ([^18^F]FDG) levels with anatomical–functional analyses (PET/CT), we found a reduced hepatic and brain glucose uptake in the AIP mice compared to WT (Figure 3A,B), thus confirming previously reported data [33,34]. Such findings were consistent with the condition of hyperinsulinemia observed in fasted AIP animals, and supported the concept that the latter are poorly capable of internalizing glucose. Dietary supplementation with *B. coagulans* did not improve [^18^F]FDG uptake in the liver (Figure 3A) and showed no effect on Glut1/2 glucose transporter and insulin receptor (InsR) β chain protein expression (Figure 3C). However, *B. coagulans* supplementation induced the expression of brain-specific Glut1 and Glut3 transporters and InsRβ chain (Figure 3D), suggesting that our intervention may modulate factors involved in insulin resistance. However, brain [^18^F]FDG levels were unchanged in the AIP mice with or without *B. coagulans* supplementation (Figure 3B).

In the skeletal muscle, the AIP mice displayed a higher, albeit nonsignificant, glucose uptake than the WT animals. *B. coagulans* supplementation significantly increased glucose uptake in the skeletal muscle, as measured ex vivo by counter activity (Figure 3E). Interestingly, higher levels of Glut1 and the insulin-sensitive Glut4 glucose transporters were observed in the striated muscle of AIP mice upon *B. coagulans* administration compared to either the WT and/or AIP mice (Figure 3F). Remarkably, [^18^F]FDG uptake in the skeletal muscles increased by 60% in AIP mice supplemented with *B. coagulans* (Figure 3G), which could be associated with the increased lean/fat ratio and reduced fat mass found in these animals.

### 2.4. Glucose Absorption in AIP Mice: The Role of White and Brown Adipose Tissues

As aforementioned, AIP mice are leaner than WT animals, probably because they exploit fat deposits more than glucidic reserves as an alternative source of energy [10,35]. In order to evaluate glucose uptake in white fat (WAT) deposits, we collected visceral WAT, which included that of mesenteric (MES), gonadal (GON), and retroperitoneal (RT) origin, as well as subcutaneous fat (SC). AIP mice showed less ability to capture [^18^F]FDG in WAT compared to WT animals (Figure 4A).

AIP mice supplemented with *B. coagulans* showed increased [^18^F]FDG uptake in WAT when compared with the AIP controls (Figure 4A), especially in GON fat pad (0.85 ± 0.44 vs. 1.3 ± 0.91; *p* < 0.001 at ANOVA), RT (0. 8± 0.24 vs. 1.78 ± 0.66; *p* < 0.001 at ANOVA) and SC (1.15 ± 0.55 vs. 1.93 ± 0.56; *p* < 0.001 at ANOVA). However, glucose internalization in MES did not reach the values obtained in the control WT mice.

Brown adipose tissue (BAT) activity has been associated with a loss of BW and improved muscle contractility, whereas its decline has been related to reduced insulin sensitivity, impaired glucose homeostasis, and T2DM development [36]. In order to investigate the role of metabolically active BAT, half of the mice (*n* = 3/group) were kept at room temperature (25 °C, RT), while the other half underwent cold exposure (4 °C) for 1 h before [^18^F]FDG injection (*n* = 3/group).

Under RT conditions, the AIP mice showed a trend of reduced glucose uptake rate compared to the WT mice (*p* = 0.07; Figure 4B,C), reaching statistical significance in those receiving the *B. coagulans*-supplemented diet, probably because they dissipated less energy under fasting and room-temperature conditions. Of note, under the cold stimulus, [^18^F]FDG levels were not changed in the WT mice compared to room-temperature conditions (Figure 4B,C). Conversely, [^18^F]FDG uptake in BAT was increased 2-fold in the AIP mice (Figure 4B). These data suggest that upon cold exposure, AIP mice need to dissipate more heat than WT mice, probably to regulate energy balance. Interestingly, AIP mice receiving *B. coagulans* supplementation showed a reduced increase in BAT glucose absorption under cold conditions compared to the control AIP mice (Figure 4B,C). These treated AIP mice showed glucose absorption levels similar to those in the WT animals (Figure 4B,C).

Concerning the expression of glucose transporters in adipose tissues, we pooled BAT and WAT protein extracts from mice exposed to RT and cold conditions (*n* = 3 mice/group). We found that the AIP mice under *B. coagulans*-supplemented diet presented higher expression of Glut1 and Glut4 proteins in BAT (Figure 4D) but not in WAT (Figure 4E), suggesting that the high [^18^F]FDG absorption in BAT was coupled with an enhanced expression of glucose transporters.

### 2.5. B. coagulans Administration Modified the Gut Microbiome and Contributed to Raising Heme Content in AIP Mice

As expected, the hepatic PBGD deficiency was coupled to reduced heme levels in the AIP mice (Figure 5A). Surprisingly, *B. coagulans* supplementation increased the hepatic heme contents in the AIP mice (*p* < 0.01 *B. coagulans* vs. AIP), restoring the hepatic heme levels to those found in the WT mice. Given that no changes were observed in the expression of the genes coding for the regulatory heme pathway enzymes (ALAS1, Peroxisome proliferator-activated receptor γ co-activator 1 α [PGC1-α], PBGD, and Heme oxygenase-1 [HO-1]; Appendix A), these data suggest that hepatic heme availability might be enhanced by an enriched gut microbiome.

As expected, upon the analysis of fecal samples, *B. coagulans* was only found in treated animals and was undetectable in the control groups (Appendix A). The diversity of fecal microbiome was significantly higher in the AIP mice fed a *B. coagulans* supplement (alpha diversity S obs: *p* = 0.003, and beta diversity, feature-level PCoA Bray–Curtis, *p* = 0.001), while the microbial community in the AIP samples taken from the control AIP mice (24-weeks-old) and treated animals at baseline (12-weeks-old) showed a nonsignificant distance from that of the control WT (Appendix A). In order to minimize false-positive species, an FDR cut-off value of q ≤ 0.01 was established. When compared to the gut microbiota in the AIP mice, a relative abundance of *Odoribacter laneus* (AB490805) and *Bacteroides faecichinchillae* (AB574480) was observed in the WT control mice and AIP mice fed *B. coagulans* (Figure 6A), whereas *Lachnoanaerobaculum umeaense* (FJ967000) and *Lachnospiraceae bacterium* (KC311366) were more abundant in the control AIP mice and their populations were reduced in both the WT mice and AIP animals fed *B. coagulans* (Figure 6B). Probiotic administration modified the relative abundance of these four species in the AIP mice, which could be associated with the metabolic improvements observed in these animals.

Finally, supplementation with *B. coagulans* maintained the relative abundance of two other species throughout the study. *Bacteroides dorei* (AB242142), detected in young AIP mice (12-weeks-old), disappeared in the feces of 24-week-old animals, whereas *Turicibacter sanguinis* (AF349724) was undetectable in young animals but showed a higher abundance in older mice (both WT and AIP). Their respective abundances at the basal timepoint were maintained in treated-AIP mice (Figure 6C).

## 3. Discussion

Changes in gut microbiome composition in metabolic disorders are well documented [36,37]. However, it is difficult to establish whether dysbiosis is specifically associated with a genetic defect or disease status, and more importantly, whether nutritional supplementation can improve disease.

A unique nutritional aspect in AIP is that a carbohydrate-rich diet (60–70% of the total calories) is recommended to prevent or avert mild symptoms [12]. In the absence of vomiting, or paralytic ileus, preventive interventions involving increased dietary carbohydrates have been proposed for either AIP prophylaxis or the management of mild attacks [10,38]. On the other hand, a high-glucose diet is reported to increase gut permeability, the translocation of bacterial products, and insulin resistance, indicating that the gut–liver axis is likely to contribute to altered glucose homeostasis in metabolic diseases [39,40,41]. Several experimental models have highlighted that these alterations in glucose metabolism may arise in hepatic porphyrias. Female rats exposed to a combination of two porphyrin genic drugs showed a downregulation of key enzymes involved in gluconeogenesis and glycogenolysis, phosphoenolpyruvate carboxykinase (PEPCK) and glycogen phosphorylase (GP), respectively [42]. Collantes and collaborators demonstrated that a genetic mouse model of AIP developed glucose intolerance and hyperinsulinemia, and aberrantly responded to caloric restriction in the absence of porphyrinogenic stimuli or heme precursor accumulation. While WT mice induced hepatic glycogen catabolism in order to re-establish glucose homeostasis, AIP animals activated ketogenesis and gluconeogenesis, leading to an increment in ketone bodies as alternative energy sources [34].

In the present work, we performed a nutritional supplementation study in these AIP mice with alive probiotic *B. coagulans*. We found that this intervention reduced hyperinsulinemia and improved glucose uptake during severe hyperglycemia. The expression of InsRβ chain protein and tissue-specific glucose transporters was increased in the brain and striated muscle of the treated animals. We also confirmed an improved glucose uptake in skeletal muscle, as measured by the average [^18^F]FDG signal. Alive probiotics also stimulate fat disposal in both BAT and WAT. Lipid breakdown may be mediated by energy burning during muscle contraction and heat dissipation by BAT, resulting in an improvement in lean/fat ratio and increasing muscle mass. These data are consistent with the high energy expenditure associated with an improvement in hyperinsulinemia and glucose metabolism in AIP mice supplemented with *B. coagulans*.

*B. coagulans* does not exist in the intestinal microbiota. However, a number of publications confirm that the administration of a probiotic containing the *B. coagulans* strain (BC30 GBI-30, 6086) can help regulate the gut microbiota, which can directly impact the gut–liver axis, improving gastrointestinal disorders related to IR [43,44,45,46,47,48,49]. Clinical studies have shown that nutritional intervention with a single *B. coagulans* strain (TCI711 probiotics isolated from apples) improves liver fat accumulation and inflammation in patients with nonalcoholic fatty liver disease (ClinicalTrials.gov, Identifier: NCT05635474).

BC30, unlike other probiotic strains, is a spore former with a protective outer layer that survives to the upper gastrointestinal tract milieu (high acid and alkaline resistance stability), exerting beneficial effects in the lower gut with antimicrobial activity, increasing immunological defenses, and improving intestinal regularity [31,50]. Indeed, stable and spore-forming *B. coagulans* can survive harsh manufacturing processes such as shear, high temperature short time (HTST), and high-pressure processing (HPP) pasteurization for long-term storage and transport, and it is easy to formulate into functional foods, beverages, and companion animal products without changes to the composition or taste [31,50]. Indeed, it is generally recognized as safe (GRAS) for human consumption by the U.S. Food and Drug Administration (FDA). Taking all this into account, *B. coagulans* is currently one of the most commonly used probiotic strains to maintain the health of intestinal bacteria or to induce changes in the intestinal microbiota, having implications at a systemic level [43,44,45,46,47,48,49,51]. The proposed mechanisms include the modulation of gut flora and amelioration of intestinal dysbiosis, an improvement in intestinal barrier function, a reduction in chronic low-grade inflammation, and the modulation of gut peptide secretion.

From the analysis of the list of differentially represented species after the nutritional intervention, we focused our attention on the two of them which were significantly elevated in the AIP mice fed *B. coagulans* and in the control WT mice, *Odoribacter laneus* and *Bacteroides faecichinchillae*. A characterization of the probiotic potential of *Odoribacter laneus* in two different murine models of obesity revealed that the reduction in circulating succinate is concomitant with an improvement in glucose tolerance and obesity-related inflammation [52]. In our AIP mice, the expression of proinflammatory genes in the WAT and BAT samples suggests that there is no inflammation associated with adipose tissue. However, oxidative stress and inflammation have been described in association with the accumulation of heme precursors and the recurrence of attacks in patients and models of hepatic porphyrias [53]. Indeed, the level of inflammatory cytokines correlated with the abundance of specific bacterial taxa, including *Lachnospiraceae bacterium* [54]. In our study, the abundance of *Lachnospiraceae bacterium* increased with the development of the disease. The *Lachnospiraceae* family are commensal with a significant role in the digestion of carbohydrates and proteins and was significantly elevated in patients with primary sclerosing cholangitis [44].

Finally, a high abundance of *Turicibacter sanguinis* was shown in fecal samples from old WT and AIP mice, but was undetectable in young or treated-AIP mice. On the contrary, *Bacteroides dorei*, which showed a relative abundance in the gut population of young AIP mice, was not detected in 24-week-old WT and AIP animals. *Bacteroides dorei* is also an abundant species in the microbiome of newborn infants [55]. So, it is possible that the relative abundance of *T. sanguinis* and *Bacteroides dorei* with time is indicative of the transition toward the adult-like microbiome. Thus, the predominance of *B. coagulans*, belonging to the Lactobacillus family, produces L + lactic acid and could be associated with the smaller presence of this species in treated-AIP mice when compared to adult AIP mice. Of note, *Bacteroides spp*. are mainly involved in the secretion of mucus and short-chain fatty acids (SCFAs) and the activation of certain pathways in the immune system [44]. *Bacteroides dorei* exhibited an increased pattern in probiotic-administered fecal samples. *B. dorei* Bacteroides, as well as other anaerobic microorganisms, ferment undigested food and other host metabolites in the large intestine to produce beneficial SCFAs [56]. SCFAs (propionate, butyrate, and acetate) are absorbed from the intestinal cavity and subsequently distributed and metabolized within host cells. Propionate is mainly metabolized in the liver, butyrate mainly provides energy, and acetate can persist in higher concentrations in the peripheral blood for a long time [57]. Intracellular butyrate and propionate both restrain histone deacetylase activities in immune cells and promote histone hyperacetylation, thereby affecting gene expression and cell differentiation, such as the downregulation of IL6 and IL12. On the other hand, higher acetate levels can increase intracellular acetyl-CoA contents, and thus also influence histone acetylation levels [58].

Heme levels are also increased in the liver of treated-AIP mice. Transporters for heme, hemin, and iron, which are linked to the vitamin B12 transport system, are strongly increased in the newborn metagenome [59]. *B. coagulans* administration has been shown to increase the levels of L-glutamate [60], which starts the tetrapyrrole biosynthesis pathway in bacteria (http://vm-trypanocyc.toulouse.inra.fr/META/NEW-IMAGE?type=PATHWAY&object=PWY-5918, accessed on 22 July 2023). It is thus tempting to speculate that increased amounts of protoheme synthesized by *B. coagulans*, or by other enteric bacteria enriched by the treatment, could be incorporated into the hepatic heme pool through portal circulation, contributing to palliating the deficiency in endogenous heme synthesis due to PBGD impairment.

In conclusion, nutritional supplements in the form of a capsule with sporulated probiotic (BC-30) orally administered for 3 months improved glucose metabolism in AIP mice by reducing hyperinsulinemia and increasing glucose uptake in insulin-sensitive tissues (especially muscle). Changes in the gut microbiota, and microbiota-derived molecules incorporated at a systemic level from portal circulation, may be involved in these responses, as well as in the mobilization of fat deposits impacting the lean/fat ratio. Our data provide a proof of concept that BC-30 probiotic, as a dietary intervention in AIP, induces relevant changes in the gut microbiota composition, reduces hyperinsulinemia, and improves glucose uptake and the mobilization of fat deposits impacting the lean/fat ratio. Further studies are needed to evaluate whether BC30 probiotics may offer a safe and cost-effective treatment for hyperinsulinemia and altered glucose metabolism in people carrying the PBGD mutation with AIP. Moreover, these interventions may help in reducing the overweight derived from high carbohydrate intake (recommended for patients with AIP) without the risk of complications associated with fasting and caloric deprivation, which can trigger acute porphyria attacks.

However, it is not easy to extrapolate these experimental results to humans. Our proof of concept was carried out under optimal dietary conditions to minimize changes in the microbiota unrelated to the study, but human diets and habits can greatly alter the gut microbiota. It is well known that high fat intake, sugary soft drinks, medications such as stomach protectants, antibiotics, the antidiabetic drug metformin, statins, laxatives, or hemin in patients with AIP are associated with decreased gastrointestinal tract microbiota diversity [61]. In contrast, the consumption of coffee, tea, and red wine, sources of antioxidants with anti-inflammatory properties, is associated with a greater diversity in the microbiota [62]. Therefore, future trials should be aimed at establishing timing and dosing regimens able to produce significant changes in human disease. In addition, BC30 administration via commercial oral capsules is easy and would be the preferred route of administration for both safety and logistic reasons.

## 4. Materials and Methods

### 4.1. Material

Freeze-dried spores of bacteria *B. coagulans* (GBI-30 6086) were kindly provided by Ganeden Inc. (Mayfield Heights, OH, USA). The spores were recovered from a controlled fermentation of a pure culture of *Bacillus coagulans* GBI-30 6086. The product was a dried powder also containing organic inulin. The alive bacteria were diluted in drinking water at a concentration of 7 mg/mL (approximately 10^8^ spores/mL). This dose was selected based on previous studies that have shown an improvement in lipid serum profiles and the prevention of hepatic steatosis in mice fed a high-fat diet [25].

### 4.2. Study Design

Three-month-old compound heterozygous C57BL/6 ^pbgdt1(neo)Uam^/^pbgdt2(neo)Uam^ (AIP) female mice with a median weight of 19 g were randomly assigned to receive BC30™ or no treatment (*n* = 6/group). Age- and weight-matched C57BL/6 wild-type (WT) mice were also included as untreated controls. BC30™ was diluted in drinking water (DW) and administered for 12 weeks. The mice were maintained on a standard diet throughout the course of observation and food consumption and weight were monitored weekly. All the analysis was performed after 14 h of fasting. The experimental protocol was approved by the Ethics Committee of the University of Navarra (CEEA 023-22), according to the European Council guidelines.

### 4.3. Gene Expression Analysis

RNA was extracted from tissues using Trizol reagent (Life Technologies-ThermoFisher Scientific, Carlsbad, CA, USA). An amount of 1 µg of total RNA was retrotranscribed with a VILO random hexamers synthesis system (Life Technologies-ThermoFisher Scientific, Carlsbad, CA, USA). Quantitative real-time PCR (qRT-PCR) was performed using iQ SYBR Green supermix in an iQ5 real-time PCR detection system (Bio-Rad, Hercules, CA, USA) and specific primers for *Alas1, Pgc-1α,* and *Ho-1* (*Alas1*, forward: 5′-CAAAGAAACCCCTCCAGCCAATGA-3′, reverse: 5′-GCTGTGTGCCGTCTGGAGTCTGTG-3’, product length: 104 bp; *Pgc-1α*, forward: 5′-GAAGTGGTGTAGCGACCAATC-3′, reverse: 5′-AATGAGGGCAATCCGTCTTCA-3′, product length: 162 bp; *Ho-1*, forward: 5′-CCAGAGTGTTCATTCGAGCA-3′, reverse: 5′-CTGCAGGGGCAGTATCTTGC-3´, product length: 116 bp). Genes involved in bridging enzymes among glycogenolysis and gluconeogenesis were also evaluated (*G6pase*, forward 5´-AACGCCTTCTATGTCCTCTTT-3′, reverse: 5′-GTTGCTGTAGTAGCTGGTGTC-3′, product length: 168 bp; *Pepck*, forward: 5′-AGCCTGCCCCAGGCAGTGAG-3′, reverse: 5′-CATGCACCCTGGGAACCTGGC-3′, product length: 339 bp). Data were normalized to β-actin housekeeping gene (*Actb,* forward: 5′-CGCGTCCACCCGCGAG-3′, reverse: 5′-CCTGGTGCCTAGGGCG-3´, product length: 125 bp) and the results were expressed as fold increase (arbitrary units, AU) according to the formula 2^ΔCt(β-Actin)−Ct(gene)^, where ΔCt represents the difference in the threshold cycle between the target and control genes. The specific primers and product length are detailed in Solares et al. [10].

### 4.4. Glucose Tolerance Test

After 14 h of fasting, a glucose tolerance test (GTT) was performed by injecting 2 g/kg intraperitoneal (i.p.) glucose overload. One week later, a second GTT study was performed following severe hyperglycemia, defined as blood glucose concentration >300 mg/dL, induced by the administration of 5 g/kg i.p. glucose in the same mice. Glycemia was measured in blood samples collected at baseline, 5, 10, 15, 20, 25, 30, 40, 50, 60, and 90 min after glucose injection with an Accu-chek glucometer (Roche Diagnostics AVIVA, Mannheim, Germany). The AUC was calculated as the total area post-i.p. 2 g/kg glucose overload or the 1st to 3rd hour post-5 g/kg i.p. injection. To assess the velocity rate of glucose reduction, we calculated the slope from the glucose peak at 30 up to 90 min after i.p. 5 g/kg glucose injection.

### 4.5. Serum and Hepatic Biochemical Measurements

Serum insulin levels were measured through a solid-phase sandwich enzyme-linked immunosorbent assay (ELISA) from Invitrogen (Waltham, MA USA). According to the manufacturer’s instructions, murine serum samples (100 µL), which were previously diluted 1:2 in diluent buffer, were first loaded onto a microplate precoated with a mouse insulin antibody and incubated at 4 °C overnight in order to allow binding (capture) by the primary antibody. Subsequently, the wells were washed four times with 300 µL of washing buffer to remove any residues. Then, 100 µL of the second (detector) biotin-conjugated antibody was added and incubated for 1 h at room temperature (RT). Horseradish peroxidase (HRP)-conjugated Streptavidin was added to the wells, and incubation continued for 45 min at RT. Next, 3,3′,5,5′-Tetramethylbenzidine (TMB) substrate was added, and a reaction with the enzyme antibody–target complex to produce measurable signal was allowed for 30 min in the dark. Finally, Stop Solution (50 µL) was added to each well, changing the solution color from blue to yellow. The absorbance at λ = 450 nm and λ = 540 nm (background) was measured using a Tecan microplate reader (Tecan Group, Switzerland). The intensity of the signal was directly proportional to the concentration of insulin present in the original specimens (λ = 450 nm). Hepatic heme was measured in frozen liver samples using a Heme Assay Kit (Abcam, Cambridge, UK).

### 4.6. Western Blot Analysis

Proteins were extracted from the liver, skeletal muscle (gastrocnemius), brain, white fat tissue (gonadal), and brown adipose tissue (BAT) using RIPA buffer containing 1 mmol/L Na-orthovanadate, 200 mmol/L phenylmethyl sulfonyl fluoride, and 0.02 μg/μL aprotinin. The samples were pooled prior to electrophoretic separation, and all reactions were performed in duplicate. Then, equal amounts of proteins (50 μg) were separated with SDS-PAGE, transferred electrophoretically to a nitrocellulose membrane (BioRad, Hercules, CA, USA), and incubated with specific antibodies overnight. At least three independent lots of freshly extracted proteins were used for the experiments. The murine antibodies exploited in this study were: anti-Glut2 (sc-518022), anti-Glut3 (sc-74497), anti-Glut1 (sc-377228), anti-Glut4 (sc-53566), anti-InsRβ (sc-57342), and anti-Vinculin (EPR8185).

### 4.7. Quantitative Magnetic Resonance (MRI) and by Micropositron Emission Tomography (MicroPET)

Mice were fasted overnight (14 h) with ad libitum access to drinking water. For body composition, in vivo measurements were performed using EchoMRI-100-700 (Echo Medical System, Houston, TX, USA) as previously described [10].

Glucose metabolism was studied in vivo via MicroPET imaging using the MicroPET studies with a glucose analog. A dose of 9.2 ± 0.9 MBq of [^18^F]FDG was injected intravenously in a tail vein. The animals were anesthetized with 2% isoflurane in 100% O_2_ gas during the uptake period of 50 min and placed prone on the scanner bed for 15 min image acquisition. A subgroup of animals were exposed to cold stimulation (4 °C) for 1 h before the [^18^F]FDG injection and also during the uptake period. Studies were performed in a small-animal-dedicated tomograph (Mosaic, Philips, Cleveland, OH, USA) followed by a CT image performed in a U-SPECT6/E-class (MILabs) scanner to obtain the corresponding anatomical image of the animals. All the studies were reconstructed applying dead time, decay, random, and scattering corrections into a 128 × 128 matrix with a 1 mm voxel size. Then, images were exported and analyzed using the PMOD v 4.105 software (PMOD Technologies Ltd., Adliswil, Switzerland) and transformed to standardized uptake value (SUV) units using the formula SUV = [tissue activity concentration (Bq/cm3)/injected dose (Bq)] × body weight (BW)(g). For a semiquantitative analysis of the images, [^18^F]FDG uptake by BAT was evaluated drawing a spherical volume-of-interest (radius = 2 mm) on PET images over the interscapular BAT (BAT), brain, heart, liver, and hindlimb’s muscle. Then, a semiautomatic segmentation was performed including the voxels with a value greater than 50% of the maximum value of the volume. Finally, the average of the SUV values within the semiautomatic VOI was calculated (SUV mean).

We conducted ex vivo [^18^F]FDG studies after dissection. At the end of the experimental period, the mice were euthanized and we collected blood samples; the liver; the kidney; the brain; the heart; skeletal muscle (gastrocnemius); white adipose tissue (WAT) fractions including gonadal (GON), retroperitoneal (RT), mesenteric (MES), and subcutaneous (SC); and the interscapular brown adipose tissue (BAT). The tissues were weighed and subjected to a radioactivity (Bq) measurement with a gamma counter (Hidex Automatic Gamma Counter). [^18^F]FDG was expressed as % of ID per gram, correcting the measurement with the calculated radioactive decay of 18-fluorine.

### 4.8. Metagenomics Analysis of the Gut Microbiome

Fecal samples were collected and frozen at −80 °C for metagenomics analysis. Fecal DNA was extracted from the stool samples and sequenced on the MiSeq platform (Illumina, San Diego, CA, USA) in CIMA Labs Diagnostics (Pamplona, Spain). Bacterial DNA isolation was carried out with Promega-Maxwell^®^ RSC equipment using the Maxwell RSC Fecal Microbiome DNA Kit (Promega Corporation, Madison, WI, USA). For each DNA sample, the V3–V4 hypervariable regions of the 16S rRNA gene were amplified using specific primers (Illumina). The 16S rRNA sequences were filtered using BaseSpace™ Sequence Hub (Illumina), and an amplicon sequence variant (ASV) abundance matrix was generated. Finally, taxonomy was assigned using a Ribosomal Database Project (RDP) classifier (v3 May 2018 DADa2 32bp).

### 4.9. Statistical Analysis

Data were log-transformed and the differences between groups were analyzed using one-way ANOVA followed by Bonferroni correction. Adjusted (adj) *p* values < 0.05 were considered statistically significant. Statistical analyses were performed using Prism software (version 9.1, GraphPad Software).

Statistical differences in gut microbiota abundances were assayed using the MicrobiomeAnalyst 2.0 platform [63]. Data were filtered by removing ASVs with <4 counts in at least 20% of samples and variance < 10% of the interquantile range. The counts of the rest of the abundance matrix were transformed to a centered log-ratio (CLR) to avoid issues related to compositional data, and the differential abundance of the four groups was analyzed using multiple linear regression. Alpha diversity at the feature level was analyzed with unfiltered data and using a parametric test. Beta diversity was analyzed using the ordination-based method principal coordinate analysis (PCoA), the Bray–Curtis distance, the statistical method PERMANOVA, and data (feature-level) normalized by relative log expression. Venn diagrams were made using the venny 2.1.0 software “https://bioinfogp.cnb.csic.es/tools/venny/ (last accessed on 1 February 2023)”.

## Figures and Tables

**Figure 1 ijms-24-11938-f001:**
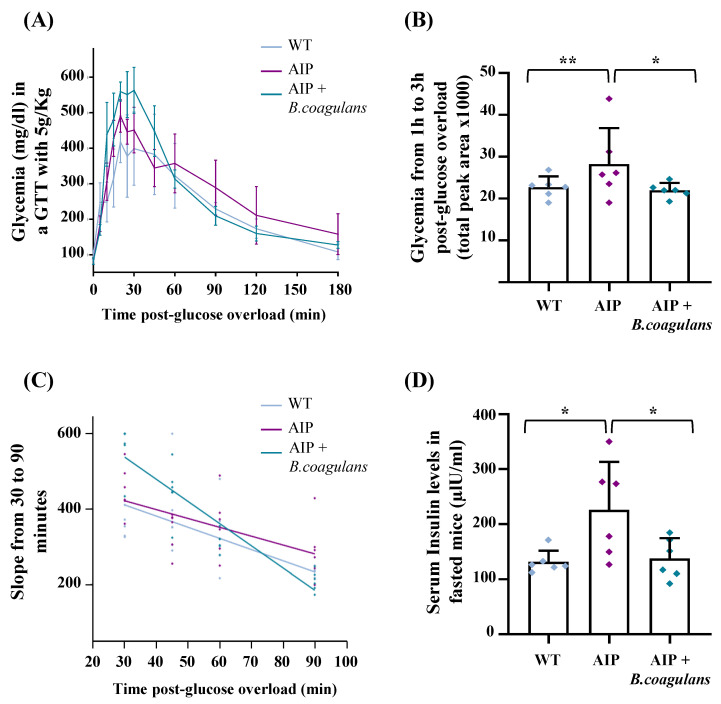
Feeding AIP mice with a *B. coagulans*-supplemented diet ameliorates glucose tolerance and hyperinsulinemia. (**A**) GTT curves performed after 5 g/kg i.p. glucose overload to induce severe hyperglycemia. Blood glucose was measured from 0 to 180 min after glucose injection. (**B**) Box plot showing the total peak area under the curve (AUC) calculated from the 1st to the 3rd hour post-glucose overload. (**C**) Analysis of the slope was carried out from the glycemic peak at 30 min up to 90 min after carbohydrate loading. (**D**) Insulin levels were measured in serum from WT and AIP mice with or without treatments. All experimental groups were fasted 14 h before the GTT and insulin measurements. Data are shown as average and standard deviation (SD). * *p* < 0.05, ** *p* < 0.01, one-way ANOVA followed by Bonferroni post-test.

**Figure 2 ijms-24-11938-f002:**
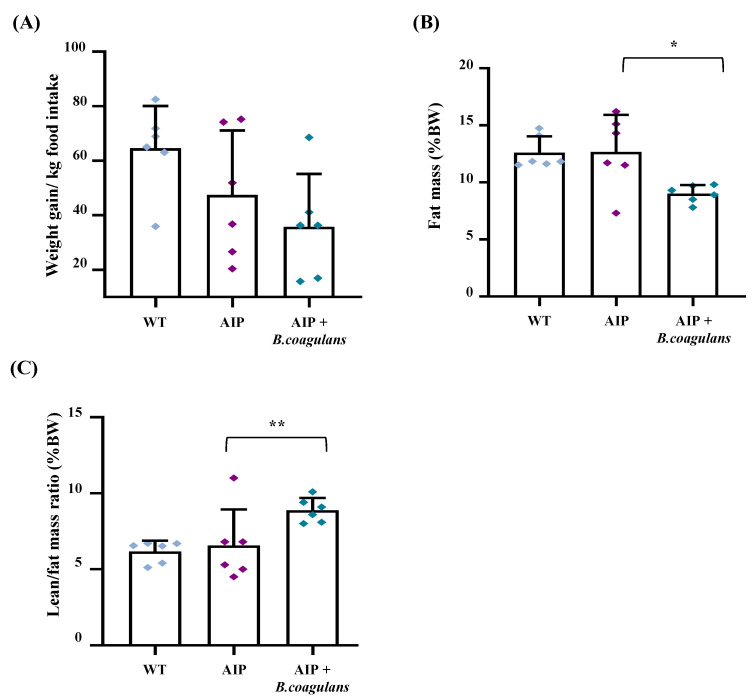
*B. coagulans*-supplemented-diet-induced weight loss was accompanied by higher lean/fat ratio. (**A**) Body weight was measured weekly in mice and normalized for kg of weekly food ingested. (**B**) Body fat mass (%) and (**C**) lean/fat mass ratio were assessed using quantitative MRI. Data are shown as mean and standard deviation (SD). * *p* < 0.05, ** *p* < 0.01, one-way ANOVA followed by Bonferroni post-test.

**Figure 3 ijms-24-11938-f003:**
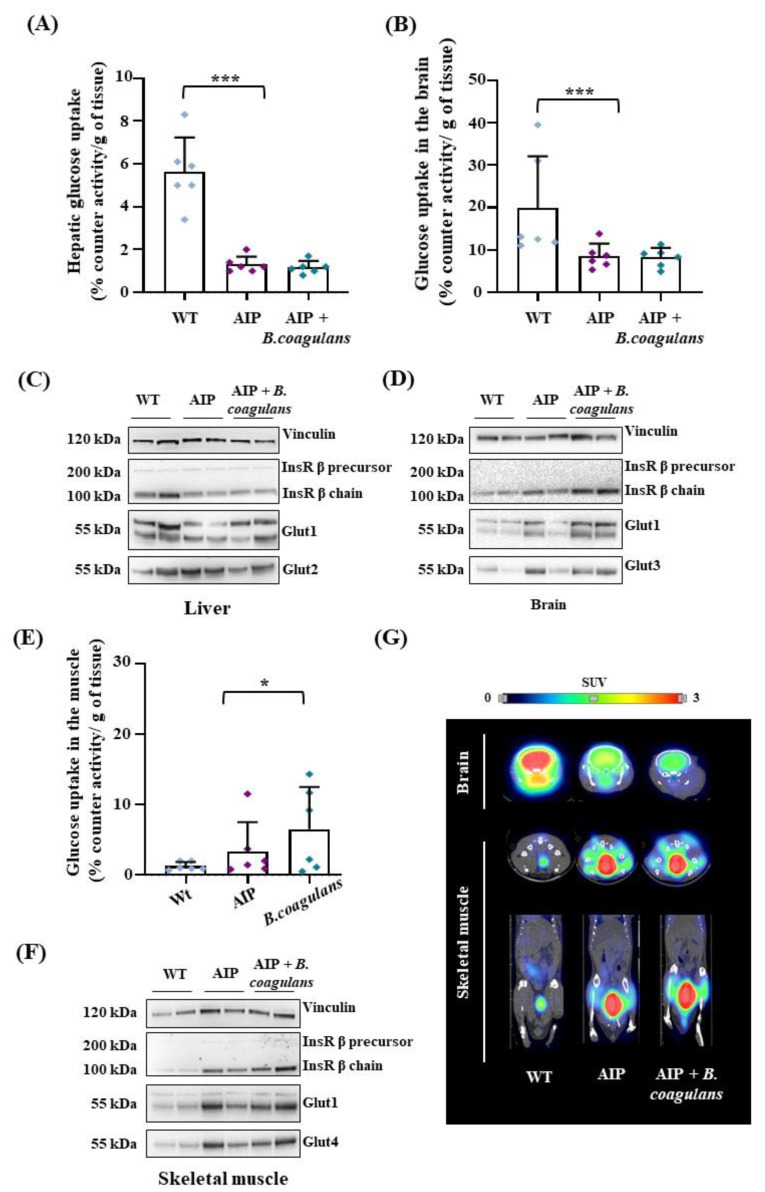
*B. coagulans* supplementation slightly modified glucose uptake in the liver and brain of AIP mice, but strongly enhanced glucose absorption in skeletal muscle. (**A**) [^18^F]FDG radioactivity measured ex vivo in mice livers. (**B**) [^18^F]FDG radioactivity measured ex vivo in the brain (right). (**C**,**D**) Glut1/2/3 and InsRβ chain analyzed using WB in liver and brain tissues. (**E**) [^18^F]FDG radioactivity measured ex vivo in skeletal muscle. (**F**) Glut1/4 and InsRβ chain analyzed usign WB in the skeletal muscle. (**G**) Representative PET/CT in vivo images of the brain (axial plane) and of skeletal muscle (tibial anterior, on top: axial plane; on bottom: coronal plane) performed 1h after [^18^F]FDG injection in fasted mice. Ex vivo acquired data were normalized by [^18^F]FDG injected volume and tissue weight (g). Values are shown as average and standard deviation (SD). * *p* < 0.05, *** *p* < 0.001, one-way ANOVA followed by Bonferroni post-test.

**Figure 4 ijms-24-11938-f004:**
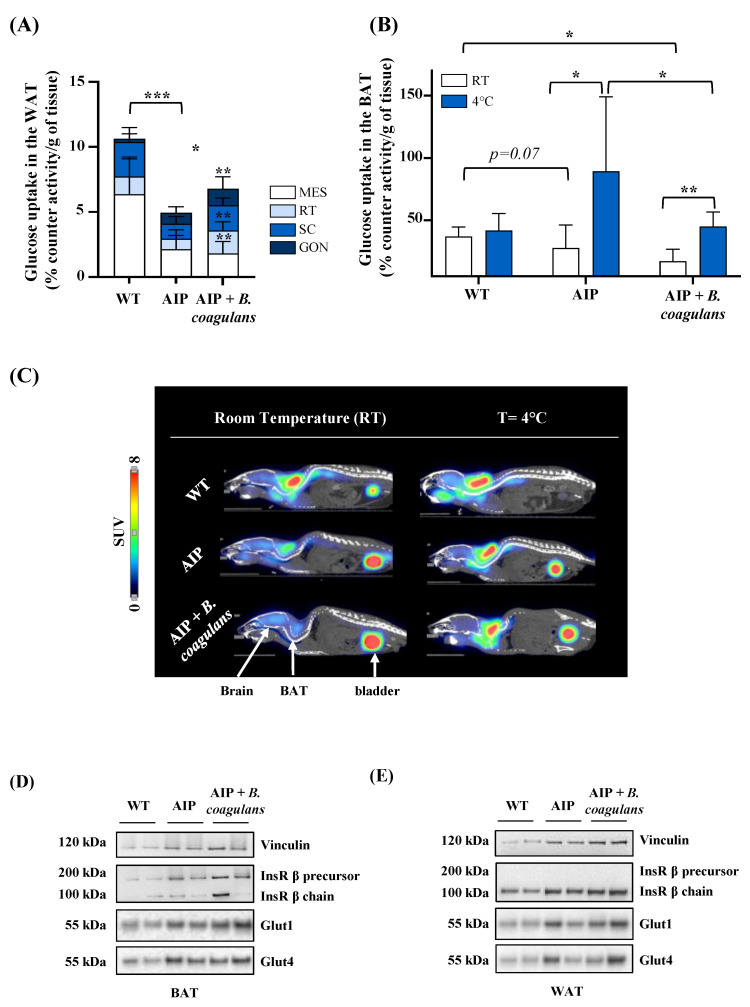
[^18^F]FDG uptake and glucose transporters in white and brown adipose tissues. (**A**) Ex vivo quantification of [^18^F]FDG radiotracer measured in white fat pads (mesenteric, gonadal, retroperitoneal, and subcutaneous). (**B**) Ex vivo quantification of [^18^F]FDG radiotracer measured in BAT at RT and after cold stimulus (left). (**C**) Representative PET/CT in vivo images of [^18^F]FDG uptake in BAT (interscapular space and sagittal plane) performed at RT and 4 °C in fasted mice. Data were normalized on [^18^F]FDG injected volume and tissue weight (g). Representative images of WBs from Glut1/4 and InsRβ chain assessed in (**D**) BAT and (**E**) WAT (gonadal fat) in mice were challenged with cold. Values are shown as mean and standard deviation (SD). * *p* < 0.05, ** *p* < 0.01, *** *p* < 0.001, one-way ANOVA followed by Bonferroni post-test.

**Figure 5 ijms-24-11938-f005:**
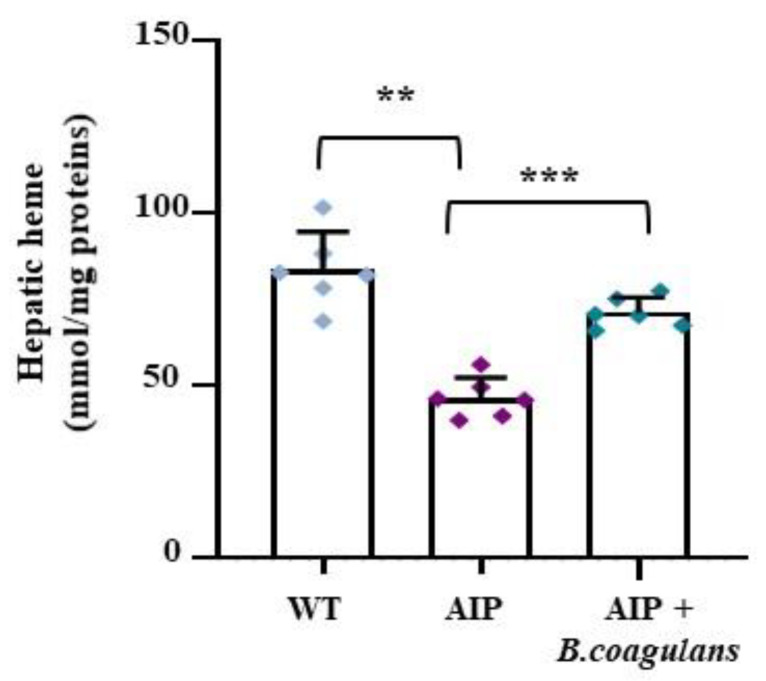
*B. coagulans* supplementation enhanced hepatic heme content in AIP mice. Bar graph shows heme content as colorimetrically measured in frozen liver samples. Data were normalized by mg of liver protein. ** *p* < 0.01, *** *p* < 0.001, one-way ANOVA followed by Bonferroni post-test.

**Figure 6 ijms-24-11938-f006:**
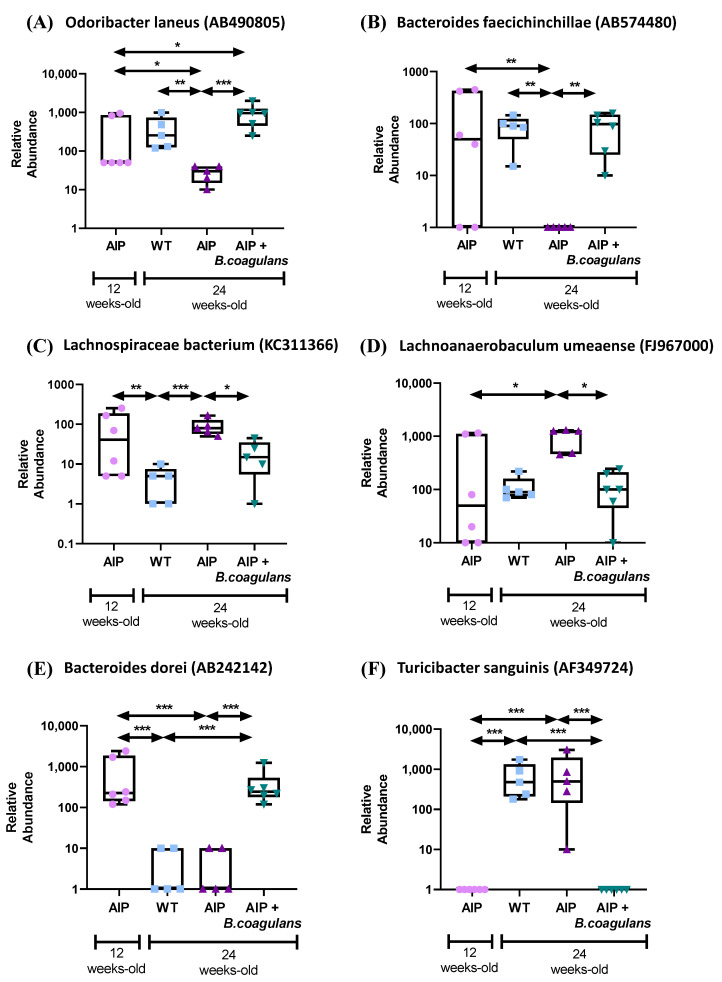
Gut microbiota changes associated with the disease and nutritional supplementation with *B. coagulans*. (**A**) Bacterial species not present or with reduced abundance in the gut microbiota of AIP mice when compared to WT that were increased after the probiotic supplementation. (**B**) Bacterial species highly represented in the gut microbiome in AIP mice, and that were restored upon *B. coagulans* administration to levels found in control WT animals. (**C**–**F**) Bacterial species that changed their presence with age in both WT and AIP mice, but nutritional supplementation with *B. coagulans* maintained their presence with respect to basal time levels. * *p* < 0.05, ** *p* < 0.01, *** *p* < 0.001, one-way ANOVA followed by Bonferroni post-test.

## Data Availability

The data presented in this study are available on request from the corresponding author.

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
