# Peer review of "Nutritional Interventions with Bacillus coagulans Improved Glucose Metabolism and Hyperinsulinemia in Mice with Acute Intermittent Porphyria"

_ijms, 2023, doi:10.3390/ijms241511938_

Round 1

Reviewer 1 Report

Comments and Suggestions for Authors

The authors describe an experimental study related to AIP mice which have used probiotics for 12 weeks (short period) changing their liver metabolism especially related to glucose metabolism.  The animal study is done carefully but the text would have benefited from explanation of liver metabolism of AIP mice compared to WT mice in general. Information and comparison of human AIP to AIP mice is confusing and should be shortened dramatically since even the authors explained at the end that it is hard to say if this has anything to do with humans.      

A mutation in the PBGD gene does not necessarily mean that an individual has acute intermittent porphyria which is a very rare disease. Many base change changes have too little effect on the enzyme activity that it would cause any harm and thus, we should not diagnose people by a mutation detection solely.  Mutation does not equal the disease.  

Neuropsychiatric symptoms listed are rare and occur only during severe and commonly prolonged attacks. The symptoms listed here are related to acute encephalopathy and commonly in PRES in severe attacks only.  More common are related to mild mental symptoms and autonomous neuropathy as listed here gastrointestinal but tachycardia and hypertension, which are typical, are missing.

Peripheral motor neuropathy and central nervous system disturbances accompanied by hypertension, chronic abdominal pain…

The sentence should be removed or rephased since it mixes symptoms and signs of an acute attack and chronic phase in few severely affected patients.

…physical exhaustion - what does it mean? …certain steroid hormones, medications, other chemicals… – this causes only confusion – omit or be more precise and explain what the mechanism of it is in general

Majority of the AIP patients lead normal life and tolerate weight reduction.   

First-line strategy to manage or to prevent acute episodes consists of hemin infusion… this is an option not the first-line but for those with recurrent attacks

Is there evidence-based medicine that carbohydrate as such would be beneficial to inhibit ALAS1 activity in clinical trials.  Patients used to even die during attacks despite carbohydrate loading. Mortality rate has been high.

In many cases, eradicating triggering factors starts healing, and thus, an attack usually lasts a week. This is also commonly related to sex-hormones in women. Premenstrum is the most common triggering factor in women, and when the bleeding starts, estrogen and progesteron levels decrease dramatically and normalizes many liver metabolism pathways which have been up-or down-regulated during premenstrum.

 I would leave out “candy therapy” here but it is important to avoid fasting by using glucose infusions if a patient is unable to eat.  It is good to keep in mind that in many other acute diseases and most likely also in AIP glucose metabolism is secondarily mildly impaired during acute crisis but only transiently. This is mainly related to inflammatory factors of hepatic origin. Diabetes mellitus is completely other thing and needs genetic background.  AIP does not cause or prevent DMT2 unless the patients’ BMI is low but like anorexia in general, many hepatic metabolic changes occur due to severe fasting.  For example, 20-30% reduction of hepatic fat occurs within a few days during fasting in humans which demonstrates dynamicity of the liver in energy metabolism. Diabetic patients do not suffer from a lack of glucose intramuscularly.

Pregnancies are usually well-tolerated and do not differ from normal pregnancies or deliveries. The patient’s primary condition (low BMI) may affect steroid hormonal production but even infertility has been rare.

Probiotics have been studied for decades but no success stories have been reported to date, only microbiota transplantation due to recurrent Clostridium difficile infection has become ”the drug of choice”.

Studies using special mice strains do not mimic humans since common immunological factors are missing. Moreover, AIP -mice do not resemble AIP -patients but mice metabolism might be interesting for those who are working with them.   B. coagulans did not seem to taste so good since even mice reduced their food intake and started rather fasting and lost fat quickly similar to humans. Loosing weight is a quick way to control glucose levels also in human diabetics.     

The problem with these probiotics is the short time of use. Commonly after 6 months differences seem to fade away.

Author Response

Reviewer 1

The authors describe an experimental study related to AIP mice which have used probiotics for 12 weeks (short period) changing their liver metabolism especially related to glucose metabolism.  The animal study is done carefully but the text would have benefited from explanation of liver metabolism of AIP mice compared to WT mice in general. Information and comparison of human AIP to AIP mice is confusing and should be shortened dramatically since even the authors explained at the end that it is hard to say if this has anything to do with humans.

AUTHORS: We thank the reviewer for his/her comments. The text has been revised accordingly.

A mutation in the PBGD gene does not necessarily mean that an individual has acute intermittent porphyria which is a very rare disease. Many base change changes have too little effect on the enzyme activity that it would cause any harm and thus, we should not diagnose people by a mutation detection solely.  Mutation does not equal the disease.  

AUTHORS: The reviewer is right. Clinical manifestation of AIP is associated with the accumulation of heme precursors that are potentially neurotoxic. We have rewritten the second sentence of the introduction to emphasize the association between clinic and accumulation of ALA and PBG.

Neuropsychiatric symptoms listed are rare and occur only during severe and commonly prolonged attacks. The symptoms listed here are related to acute encephalopathy and commonly in PRES in severe attacks only.  More common are related to mild mental symptoms and autonomous neuropathy as listed here gastrointestinal but tachycardia and hypertension, which are typical, are missing.

Peripheral motor neuropathy and central nervous system disturbances accompanied by hypertension, chronic abdominal pain…

The sentence should be removed or rephased since it mixes symptoms and signs of an acute attack and chronic phase in few severely affected patients.

AUTHORS: We have changed the sentence according to your suggestion.

…physical exhaustion - what does it mean? …certain steroid hormones, medications, other chemicals… – this causes only confusion – omit or be more precise and explain what the mechanism of it is in general

Majority of the AIP patients lead normal life and tolerate weight reduction.   

First-line strategy to manage or to prevent acute episodes consists of hemin infusion… this is an option not the first-line but for those with recurrent attacks

AUTHORS: We thank the reviewer for this comment. These concepts have now been included in the Introduction.

Is there evidence-based medicine that carbohydrate as such would be beneficial to inhibit ALAS1 activity in clinical trials.  Patients used to even die during attacks despite carbohydrate loading. Mortality rate has been high.

In many cases, eradicating triggering factors starts healing, and thus, an attack usually lasts a week. This is also commonly related to sex-hormones in women. Premenstrum is the most common triggering factor in women, and when the bleeding starts, estrogen and progesteron levels decrease dramatically and normalizes many liver metabolism pathways which have been up-or down-regulated during premenstrum.

 I would leave out “candy therapy” here but it is important to avoid fasting by using glucose infusions if a patient is unable to eat.  It is good to keep in mind that in many other acute diseases and most likely also in AIP glucose metabolism is secondarily mildly impaired during acute crisis but only transiently. This is mainly related to inflammatory factors of hepatic origin. Diabetes mellitus is completely other thing and needs genetic background.  AIP does not cause or prevent DMT2 unless the patients’ BMI is low but like anorexia in general, many hepatic metabolic changes occur due to severe fasting.  For example, 20-30% reduction of hepatic fat occurs within a few days during fasting in humans which demonstrates dynamicity of the liver in energy metabolism. Diabetic patients do not suffer from a lack of glucose intramuscularly.

Pregnancies are usually well-tolerated and do not differ from normal pregnancies or deliveries. The patient’s primary condition (low BMI) may affect steroid hormonal production but even infertility has been rare.

AUTHORS: These points have been amended in the new version of the manuscript.

Probiotics have been studied for decades but no success stories have been reported to date, only microbiota transplantation due to recurrent Clostridium difficile infection has become ”the drug of choice”.

AUTHORS: Probiotics are generally safe and well-tolerated. Accumulating evidences suggest its potential to improve insulin sensitivity and blood sugar control in a cost-effective manner.

Studies using special mice strains do not mimic humans since common immunological factors are missing. Moreover, AIP -mice do not resemble AIP -patients but mice metabolism might be interesting for those who are working with them.  B. coagulans did not seem to taste so good since even mice reduced their food intake and started rather fasting and lost fat quickly similar to humans. Loosing weight is a quick way to control glucose levels also in human diabetics.     

AUTHORS: Spores of B. coagulans were administered in the drinking water, however, these mice showed less food consumption, therefore a satiety effect cannot be ruled out. Of interest, BC30TM is commercialized (Ganeden BC30) and is fit for fortification of foods and beverages, such as juices, smoothies, teas, coffees, frozen foods and desserts that can improve palatability of this diet.

The problem with these probiotics is the short time of use. Commonly after 6 months differences seem to fade away.

AUTHORS: Thus, intermittent probiotics use for a medium- short-term could be an option that can be considered to periodically change intestinal bacteria microbiota composition in order to improve insulin sensitivity and blood sugar control.

Reviewer 2 Report

Comments and Suggestions for Authors

Dear Authors, 

The manuscript addresses the problem of acute intermittent porphyria (AIP) and related insulin resistance. The studies reported here are designed to evaluate the beneficial effects of nutritional interventions on correcting carbohydrate dysfunction in a mouse model. The Authors observed that the addition of B. coagulans to the drinking water of mice influenced the modification of their gastrointestinal microbiome, improved glucose tolerance and hyperinsulinemia.

The Manuscript, although well written, needs the following minor corrections:

- please list keywords in alphabetical order

- line 107 "Bacillus" please write in italics

- "ex vivo, in vivo, in vitro" please write without italics as recommended by MDPI (e.g. line 186). Please correct throughout the Manuscript.

- line 378 - "Lachnospiraceae" please write in italics

- line 439 - how do the Authors know that only spores were used for the research? How can you be sure that there were no vegetative forms? how was it verified? Please clarify the information provided.

- line 458 - please provide the sequences of the primers used

- chapter 4.5 - the given method requires further development, extension with details. Please describe the ELISA in more detail, e.g. what antibodies were used, what positive control the Authors used.

- please add short, concise, concluding "Conclusions".

Best regards

Author Response

Reviewer 2

Dear Authors, 

The manuscript addresses the problem of acute intermittent porphyria (AIP) and related insulin resistance. The studies reported here are designed to evaluate the beneficial effects of nutritional interventions on correcting carbohydrate dysfunction in a mouse model. The Authors observed that the addition of B. coagulans to the drinking water of mice influenced the modification of their gastrointestinal microbiome, improved glucose tolerance and hyperinsulinemia.

The Manuscript, although well written, needs the following minor corrections:

- please list keywords in alphabetical order

AUTHORS: Done

- line 107 "Bacillus" please write in italics

AUTHORS: Done

- "ex vivo, in vivo, in vitro" please write without italics as recommended by MDPI (e.g. line 186). Please correct throughout the Manuscript.

AUTHORS: We have modified the manuscript accordingly.

- line 378 - "Lachnospiraceae" please write in italics

AUTHORS: We have changed the sentence according to your suggestion.

- line 439 - how do the Authors know that only spores were used for the research? How can you be sure that there were no vegetative forms? how was it verified? Please clarify the information provided.

AUTHORS: BC30TM is a spore-forming probiotic, unlike most other probiotics on the marked which are vegetative cell. Backed by over 25 published papers, clinical trials and approved by U.S. Food and Drug Administration (FDA) can help support digestive health, and support protein utilization. Spores contain a naturally-protective outer layer, that allows BC30TM to survive most processing conditions, including the extremes of pH, heat, cold and pressure, and transition through the gut to promote healthy bacteria. BC30TM is a natural probiotic ingredient used by product manufacturers to create functional foods and beverages.

The information on the spore count was obtained from the specifications provided to us by the company.

- line 458 - please provide the sequences of the primers used.

AUTHORS: We have provided the sequences of the primers used.

- chapter 4.5 - the given method requires further development, extension with details. Please describe the ELISA in more detail, e.g. what antibodies were used, what positive control the Authors used.

AUTHORS: The ELISA was commercially available from Invitrogen, which provides information about how to perform the assay, how to dilute samples and reagents and so on, but they do not specify the details of the reagent used. We have added a few more details from the datasheet method.

- please add short, concise, concluding "Conclusions".

AUTHORS: According to the reviewer's suggestion, mistakes/typos have been corrected and a short and concise conclusion have been included. Information about spore-forming probiotic BC30TM, the primer used and ELISA test were provided when that information was available.

We thank the reviewer for these comments which allowed us to improve our manuscript.